# Ecological Function Analysis and Optimization of a Recompression S-CO_2_ Cycle for Gas Turbine Waste Heat Recovery

**DOI:** 10.3390/e24050732

**Published:** 2022-05-21

**Authors:** Qinglong Jin, Shaojun Xia, Tianchao Xie

**Affiliations:** College of Power Engineering, Naval University of Engineering, Wuhan 430033, China; 18404133@masu.edu.cn (Q.J.); tianchaoxie94@gmail.com (T.X.)

**Keywords:** recompression S-CO_2_ Brayton cycle, finite-time thermodynamics, neural network prediction, ecological function

## Abstract

In this paper, a recompression S-CO_2_ Brayton cycle model that considers the finite-temperature difference heat transfer between the heat source and the working fluid, irreversible compression, expansion, and other irreversibility is established. First, the ecological function is analyzed. Then the mass flow rate, pressure ratio, diversion coefficient, and the heat conductance distribution ratios (HCDRs) of four heat exchangers (HEXs) are chosen as variables to optimize cycle performance, and the problem of long optimization time is solved by building a neural network prediction model. The results show that when the mass flow rate is small, the pressure ratio, the HCDRs of heater, and high temperature regenerator are the main influencing factors of the ecological function; when the mass flow rate is large, the influences of the re-compressor, the HCDRs of low temperature regenerator, and cooler on the ecological function increase; reasonable adjustment of the HCDRs of four HEXs can make the cycle performance better, but mass flow rate plays a more important role; the ecological function can be increased by 12.13%, 31.52%, 52.2%, 93.26%, and 96.99% compared with the initial design point after one-, two-, three-, four- and five-time optimizations, respectively.

## 1. Introduction

A gas turbine (GT) has the advantages of a compact structure, small volume, light weight, high power density, and high thermal efficiency and is mostly used as a driving machine in surface ships and civilian ships. However, its waste heat temperature is high and has the possibility of re-utilization. Strengthening the utilization of this part of the waste heat can effectively improve the energy utilization rate [1]. The characteristics of high efficiency, better stability, economy, compactness, and simplicity make the Brayton cycle (BC) system based on supercritical carbon dioxide (S-CO_2_) have broad application prospects [2,3,4,5,6]. Moreover, the optimal heat source temperature range of the S-CO_2_ Brayton cycle (SCBC) is 450~700 °C [1], which matches the exhaust gas temperature of GT. Therefore, it is an ideal cycle for recovering the waste heat of GTs [7,8,9,10,11].

Sulaiman et al. [12] analyzed the performances of five different types of SCBCs integrated with solar thermal power plants, including regenerative, recompression, etc. The results show that the recompression S-CO_2_ Brayton cycle (RCSCBC) has the highest thermal efficiency. Vasquez et al. [13] analyzed the regenerative, recompressed, partially cooled, and intercooled SCBC integrated with the solar receiver and obtained the optimal operating conditions through multi-objective thermodynamic optimization. Anton et al. [14] used thermodynamic modeling and analysis based on the combination of the RCSCBC with a sodium-cooled fast reactor to explore the effect of different structural layouts on the thermal efficiency of the cycle. Alharbi et al. [15] proposed a multi-effect desalination system driven by RCSCBC waste heat, which can be used for electricity and fresh water production.

Liu et al. [16] carried out the design and analysis of the RCSCBC for ships with an output power of 40MW. The research results showed that the designed RCSCBC reached an efficiency of 45.06% at 823.15 K, which meets the design requirements and improves by 8.28% over the simple regenerative cycle. Li et al. [17] conducted a comparative study on SCBC for waste heat recovery from GTs, where selected cycles were optimized and analyzed in terms of system efficiency, etc. Khadse et al. [18] conducted a thermodynamic optimization study of the RCSCBC for waste heat recovery applications; the results show that higher pressures can result in a higher cycle power output.

Mohammadi et al. [19] proposed a combined cycle consisting of a GT cycle, an RCSCBC, and an organic Rankine cycle. The research results showed that the cycle could effectively improve the utilization rate of heat recovery. Saeed and Kim [20] proposed a novel SCBC, which consisted of four compression processes and one expansion process, with a better ability to integrate with heat sources by comparing with the regenerative, recompression, and intercooling and regenerative cycles, and the results showed that the thermal efficiency (TEF) was comparable to that of the intercooling and regenerative cycle. Khatoon et al. [21] investigated the thermodynamic performance of an RCSCBC integrated with a direct air-cooled heat exchanger (HEX), and the research results showed that the cooling process of S-CO_2_ was sensible heat transfer, and the thermophysical properties changed nonlinearly.

Although the above work has carried out analysis and research on different types of SCBCs, these studies have not considered in detail the effects of irreversible factors such as irreversible compression and irreversible expansion. The overall optimum performances of the SCBC device have not been obtained under the condition of a constant overall size of the heat exchange devices.

Finite-time thermodynamics (FTT) [22,23,24,25,26,27] realizes the cross-integration of multiple basic discipline theories including thermodynamics, heat transfer, and fluid mechanics and is an important branch of modern thermodynamic theory. FTT seeks more practical and useful performance bounds and obtains the optimal performance under limited size constraints and the optimal way to achieve the goal [28,29,30,31], whose results have important engineering value. FTT has been used in the studies of a wide variety of processes and cycles, such as the organic Rankine cycle [32,33,34], Carnot cycle [35], thermoelectric generator [36], Kalina cycle [37,38], and Stirling heat engine [39,40]. Combining the analysis method of FTT theory with the SCBC, on the one hand, makes up for the deficiency of the traditional pure thermodynamic analysis method in the irreversibility analysis of the SCBC. On the other hand, it can further expand the scope of application research objects of FTT, which has important theoretical and engineering significances.

In Brayton cycle research based on the “ideal gas hypothesis”, Cheng and Chen [41,42,43] established a Brayton cycle model by considering the irreversible losses of the cyclic processes; derived the functional expressions of the cyclic power, efficiency, and ecological functions; and optimized the cycle with these performance objectives. Ust et al. [44] analyzed and studied the irreversible closed Brayton cycle with a constant-temperature heat source and the ecological performance coefficient. Kaushik et al. [45] studied the closed variable-temperature irreversible regenerative Brayton cycle model, aiming at cycle power and efficiency. Tyagi et al. [46,47,48,49] established a closed irreversible intercooling regenerative Brayton cycle model with constant-temperature and variable-temperature heat sources, respectively, and analyzed the power, efficiency, and ecological performance coefficient of the cycle.

In the FTT study of the SCBC, Na et al. [50] studied a preheating SCBC and investigated the influences of related parameters on thermal efficiency and net power output. Jin et al. [51] established an FTT model of the regenerative S-CO_2_ Brayton cycle and carried out multi-objective optimization.

An RCSCBC model that considers the finite temperature difference heat transfer between the heat source and the working fluid (WF), irreversible compression, and irreversible expansion will be established in this paper. Firstly, the ecological functions are analyzed; then, under the constraint of the total HEX inventory, taking the mass flow rate, pressure ratio, diversion coefficient, and the heat conductance of each HEXs as optimization variables, performance optimization for ecological function will be performed; and the optimal design results with practical engineering application value will be given.

## 2. Physical Model

Figure 1 and Figure 2 are the RCSCBC device diagram and *T*-s diagram, respectively. The RCSCBC device is mainly composed of the main compressor, re-compressor, low temperature regenerator (LTR), high temperature regenerator (HTR), heater, turbine, and cooler. In Figure 2, 1-2*_S_* and 1-2 denote the ideal reversible and the actual irreversible adiabatic compression processes in the main compressor, respectively; 2-8 and 8-3 denote the heat absorption process in the LTR and HTR, respectively; 3-4 denotes the heat absorption process from the high-temperature heat reservoir; 4-5 and 4-5*_S_* denote the actual irreversible and the ideal reversible adiabatic expansion processes, respectively; 5-6 and 6-7 denote the heat release processes in the LTR and HTR. At state point 7, the WF is divided by a flow divider, and a part of the WF enters the re-compressor; 8-7 is the actual irreversible adiabatic compression process in the re-compressor. After this, part of the WF leaves the re-compressor; it reaches the temperature of state point 8 after an isenthalpic mixing process with the WF after endothermic heating in the LTR, and 7-1 indicates the exothermic process.

We regarded $m_{\mathrm{wf}}$ as the mass flow rate of the WF; $m_{\mathrm{wf},1}$ as the mass flow rate of WF entering the cooler through the splitter after the S-CO_2_ comes out of the LTR; and $m_{\mathrm{wf},2}$ as the mass flow rate of WF entering the re-compressor through the splitter. Defining the ratio of $m_{\mathrm{wf},1}$ to $m_{\mathrm{wf}}$ as the diversion coefficient *x*_p_, there is the following formula:(1)$$m_{\mathrm{wf},1}=x_{p}\cdot m_{\mathrm{wf}}$$
(2)$$m_{\mathrm{wf},2}=\left(1-x_{p}\right)\cdot m_{\mathrm{wf}}$$

The efficiencies of main compressor, re-compressor, and turbine are represented by $\eta_{c}$, $\eta_{c,2}$, and $\eta_{t}$:(3)$$\eta_{c}=\left(h_{2s}-h_{1}\right)/\left(h_{2}-h_{1}\right)$$
(4)$$\eta_{c,2}=\left(h_{8s}-h_{7}\right)/\left(h_{8,\mathrm{RC}}-h_{7}\right)$$
(5)$$\eta_{t}=\left(h_{4}-h_{5}\right)/\left(h_{4}-h_{5s}\right)$$
where $h_{1}$, $h_{2s}$, $h_{8,\mathrm{LTR}}$, $h_{8,\mathrm{RC}}$, $h_{8}$, $h_{2}$, $h_{3}$, $h_{4}$, $h_{5s}$, $h_{5}$, $h_{6}$, $h_{8s}$, and $h_{7}$ are the specific enthalpies of the corresponding state points.

π is the pressure ratio. $T_{H,\mathrm{in}}$, $T_{H,\mathrm{out}}$, $T_{L,\mathrm{in}}$, and $T_{L,\mathrm{out}}$ are the inlet and outlet temperatures of the hot and cold source, respectively. $T_{8,\mathrm{LTR}}$ is the temperature of the WF after being heated by LTR, $T_{8,\mathrm{RC}}$ is the temperature of S-CO_2_ after leaving the re-compressor, and $T_{8}$ is the temperature of WF after the isenthalpic mixing process; $T_{0}$ is the ambient temperature. $m_{H}$ and $m_{L}$ are the mass flow rates of the heat source and the cold source. The heat conductances of the heater, the cooler, the LTR, and the HTR are expressed as $U_{H}$, $U_{L}$, $U_{\mathrm{LTR}}$, and $U_{\mathrm{HTR}}$, respectively. According to the HEX theory, the heat absorption rate $Q_{H}$, heat release rate $Q_{L}$, and heat recovery rates $Q_{\mathrm{LTR}}$ and $Q_{\mathrm{HTR}}$ are, respectively, given by [52,53,54,55,56]:(6)$$Q_{H}=U_{H}\cdot \frac{\left(T_{H,\mathrm{in}}-T_{4}\right)-\left(T_{H,\mathrm{out}}-T_{3}\right)}{\ln\left[\left(T_{H,\mathrm{in}}-T_{4}\right)/\left(T_{H,\mathrm{out}}-T_{3}\right)\right]}=c_{p,H}\cdot m_{H}\cdot \left(T_{H,\mathrm{in}}-T_{H,\mathrm{out}}\right)$$
(7)$$Q_{L}=U_{L}\cdot \frac{\left(T_{7}-T_{L,\mathrm{out}}\right)-\left(T_{1}-T_{L,\mathrm{in}}\right)}{\ln\left[\left(T_{7}-T_{L,out}\right)/\left(T_{1}-T_{L,\mathrm{in}}\right)\right]}=c_{p,L}\cdot m_{L}\cdot \left(T_{L,\mathrm{out}}-T_{L,\mathrm{in}}\right)$$
(8)$$Q_{\mathrm{LTR}}=U_{\mathrm{LTR}}\cdot \frac{\left(T_{6}-T_{8,LTR}\right)-\left(T_{7}-T_{2}\right)}{\ln\left[\left(T_{6}-T_{8,\mathrm{LTR}}\right)/\left(T_{7}-T_{2}\right)\right]}$$
(9)$$Q_{\mathrm{HTR}}=U_{\mathrm{HTR}}\cdot \frac{\left(T_{5}-T_{3}\right)-\left(T_{6}-T_{8}\right)}{\ln\left[\left(T_{5}-T_{3}\right)/\left(T_{6}-T_{8}\right)\right]}$$

From the thermal properties of the WF, $Q_{H}$, $Q_{L}$, $Q_{\mathrm{HTR}}$, and $Q_{\mathrm{LTR}}$ are, respectively,
(10)$$Q_{H}=m_{\mathrm{wf}}\cdot \left(h_{4}-h_{3}\right)$$
(11)$$Q_{L}=m_{\mathrm{wf},1}\cdot \left(h_{7}-h_{1}\right)$$
(12)$$h_{8}\cdot m_{\mathrm{wf}}=h_{8,\mathrm{LTR}}\cdot m_{\mathrm{wf},1}+h_{8,\mathrm{RC}}\cdot m_{\mathrm{wf},2}$$
(13)$$Q_{\mathrm{HTR}}=m_{\mathrm{wf}}\cdot (h_{3}-h_{8})=m_{\mathrm{wf}}\cdot (h_{5}-h_{6})$$
(14)$$Q_{\mathrm{LTR}}=m_{\mathrm{wf},1}\cdot \left(h_{8,LTR}-h_{2}\right)=m_{\mathrm{wf}}\cdot \left(h_{6}-h_{7}\right)$$

The total HEX inventory $U_{T}$ is the sum of the $U_{H}$, $U_{L}$, $U_{\mathrm{LTR}}$, and $U_{\mathrm{HTR}}$.
(15)$$U_{T}=U_{\mathrm{LTR}}+U_{\mathrm{HTR}}+U_{L}+U_{H}$$

Define the heat conductance distribution ratio ψ. The ψ of the heater, the cooler, LTR, and HTR are expressed as $\psi_{H}=U_{H}/U_{T}$,$\psi_{L}=U_{L}/U_{T}$, $\psi_{\mathrm{LTR}}=U_{\mathrm{LTR}}/U_{T}$, and $\psi_{\mathrm{HTR}}=U_{\mathrm{HTR}}/U_{T}$, respectively. The $\psi_{H}$, $\psi_{L}$, $\psi_{\mathrm{LTR}}$, and $\psi_{\mathrm{HTR}}$ have the following relationship:(16)$$\psi_{H}+\psi_{\mathrm{LTR}}+\psi_{L}+\psi_{\mathrm{HTR}}=1$$

For the RCSCBC, the $W_{\mathrm{net}}$ is the difference between the turbine power $W_{t}$ and the compressor power consumption, $W_{c}$, and the compressor power consumption is the sum of the main compressor power consumption $W_{c,1}$ and re-compressor power consumption $W_{c,2}$. They are given by
(17)$$W_{c,1}=m_{\mathrm{wf},1}\cdot \left(h_{2}-h_{1}\right)$$
(18)$$W_{c,2}=m_{\mathrm{wf},2}\cdot \left(h_{8,\mathrm{RC}}-h_{7}\right)$$
(19)$$W_{c}=W_{c,1}+W_{c,2}$$
(20)$$W_{t}=m_{\mathrm{wf}}\cdot \left(h_{4}-h_{5}\right)$$
(21)$$W_{\mathrm{net}}=W_{t}-W_{c}$$

The entropy production rate and ecological function *E* of the cycle are:(22)$$s_{g}=m_{L}\cdot c_{p,L}\cdot \ln\left(\frac{T_{L,\mathrm{out}}}{T_{L,\mathrm{in}}}\right)-m_{H}\cdot c_{p,H}\cdot \ln\left(\frac{T_{H,\mathrm{in}}}{T_{H,\mathrm{out}}}\right)$$
(23)$$E=W_{\mathrm{net}}-T_{0}\cdot s_{g}$$

Since S-CO_2_ is an actual gas, the *p*-*V*-*T* (*V* is the volume) relationship does not satisfy the ideal gas equation of state. In addition, the relationship between the specific heat capacity of S-CO_2_, pressure, and temperature is very complicated, so it is impossible to obtain the explanatory formula for the temperature and specific enthalpy at each state point of the cycle. Numerical solutions for temperature and specific enthalpy can only be obtained programmatically. Under the given boundary conditions ($T_{H,\mathrm{in}}$, $T_{L,\mathrm{in}}$, $U_{H}$, $U_{L}$, $U_{\mathrm{LTR}}$, $U_{\mathrm{HTR}}$, $\eta_{c}$, $\eta_{c,2}$, and $\eta_{t}$), the nonlinear equation system composed of the Equations (1)–(16) is solved by the refpropm function and the @fsolve function, and the temperature and specific enthalpy of each state point can be obtained. The calculated specific enthalpy is further imported into Equations (17)–(23) to solve the NPO and TEF under the given boundary conditions. The specific calculation flow chart is shown in Figure 3. The cycle calculation program is written by MATLAB software, and the physical properties of S-CO_2_ were calculated by REFPROP [57].

According to the Ref. [50], Table 1 gives the initial design point (IDP) parameters.

Firstly, the cycle performance is analyzed, and the relationship between the *E*, *x*_p_, and π are studied and analyzed under the conditions of different $m_{\mathrm{wf}}$, $\eta_{c}$, and $\eta_{t}$. Then, with the goal of maximizing *E*, under the constraint that the *U*_T_ is a fixed value, the optimization will be carried out with the $m_{\mathrm{wf}}$, *x*_p_, π, $\psi_{H}$, $\psi_{L}$, $\psi_{\mathrm{LTR}}$, and $\psi_{\mathrm{HTR}}$ as the optimization variables.

## 3. Results and Discussion

### 3.1. Ecological Function Analysis of RCSCBC

Let $\psi_{\mathrm{LTR}}$ = 0.1, $\psi_{\mathrm{HTR}}$ = 0.2, $\psi_{H}$ = 0.4, $\psi_{L}$ = 0.3, and π = 3, respectively. Figure 4 shows the variation trend of *E* with the *x*_p_. Since the physical properties of CO_2_ change drastically near the critical point, it is best to keep the *T*_1_ above the critical point. In the case of given other parameters, the effect of the *x*_p_ on the *T*_1_ is very obvious. Figure 4 shows the minimum *x*_p_ that can be selected according to the criterion that the *T*_1_ must be higher than the critical temperature under different *m*_wf_. When the *m*_wf_ is 120 kg·s^−1^, the selectable minimum *x*_p_ is 0.718, and when the *m*_wf_ is 130 kg·s^−1^, the selectable minimum *x*_p_ is 0.76. When the *m*_wf_ is 140 kg·s^−1^, the selectable minimum *x*_p_ is 0.808. Within the value range of 0.45 < *x*_p_ < 1, the *E* increases first and then decreases with the *x*_p_. After exceeding the marked minimum *x*_p_, the *E* shows a decreasing trend. The *x*_p_ and *m*_wf_ are not just isolated variables; there is also an interaction between them. Further analysis of the regulation of the curve in Figure 4 showed that the smaller *m*_wf_, the smaller the minimum *x*_p_ of the *T*_1_ exceeding the critical temperature. This indicates that the WF split affects the temperature of each state point of the cycle, and the increase of *m*_wf_ can expand the selection range of the *x*_p_.

Let $\psi_{\mathrm{LTR}}$ = 0.1, $\psi_{\mathrm{HTR}}$ = 0.2, $\psi_{H}$ = 0.4, $\psi_{L}$ = 0.3, and *x*_p_ = 0.8, respectively. Figure 5 shows the variation trend of the *E* with the π under different $m_{\mathrm{wf}}$s. Under the same $m_{\mathrm{wf}}$, the *E* increases first and then decreases with the increase of the π. Additionally, there are the following characteristics, that is, with the increase of the $m_{\mathrm{wf}}$, the optimal π corresponding to the maximum *E* gradually decreases. This is due to that the π affects the size of the net power output in both indirect and direct processes, and this influence will change with the change of $m_{\mathrm{wf}}$ and act on the specific value of *E*. In practical engineering, an appropriate π should be selected according to the corresponding environment to achieve better ecological performance.

Let $\psi_{\mathrm{LTR}}$ = 0.1, $\psi_{\mathrm{HTR}}$ = 0.2, $\psi_{H}$ = 0.4, $\psi_{L}$ = 0.3, and *x*_p_ = 0.8, respectively. Figure 6 shows the variation trend of the *E* with the π under different $\eta_{t}$s and $\eta_{c}$s. With the increase of $\eta_{t}$ and $\eta_{c}$, the maximum *E* of the cycle increases, and the corresponding optimal π also increases gradually. This is due to that the values of $\eta_{t}$ and $\eta_{c}$ reflect the irreversibility of the expansion and compression processes. The *E*, as a thermodynamic index that compromises the *s*_g_ and *W*_net_, can measure the irreversibility of the cycle to a certain extent; that is, there is a correlation between $\eta_{t}$, $\eta_{c}$, and the *E*. Therefore, the higher the $\eta_{t}$ and $\eta_{c}$, the smaller the energy loss caused by the irreversibility of the cycle, and the larger the *E* value.

Let $\psi_{\mathrm{LTR}}$ = 0.1, $\psi_{\mathrm{HTR}}$ = 0.2, $\psi_{H}$ = 0.4, $\psi_{L}$ = 0.3, and *x*_p_ = 0.8, respectively. Figure 7 shows the three-dimensional relationship between the *E*, $m_{\mathrm{wf}}$, and π. From Figure 7, *E* decreases gradually with the increase of the $m_{\mathrm{wf}}$ and increases first and then decreases with the increase of the π. The spherical point in Figure 7 is the maximum *E* when the *T*_1_ exceeds the critical temperature.

Let *m*_wf_ = 140 kg·s^−1^, $\psi_{\mathrm{LTR}}$ = 0.1, $\psi_{\mathrm{HTR}}$ = 0.2, $\psi_{H}$ = 0.4, and $\psi_{L}$ = 0.3, respectively. Figure 8 shows the three-dimensional relationship between the *E*, *x*_p_, and π. The spherical point in Figure 8 is the maximum *E* when the *T*_1_ exceeds the critical temperature. It can be seen that the extreme points in Figure 7 and Figure 8 are far away from the actual maximum value point. This is due to that the π, *m*_wf_, and *x*_p_ have great influences on the temperature at each state point of the cycle, and the *T*_1_ will directly affect the stability of the cycle operation. Under the combined action of these factors, the value range of the *E* is not the entire surface, so the optimal value that can be selected is not the actual maximum value.

Let *m*_wf_ = 140 kg·s^−1^, *x*_p_ = 0.8, π = 3, and $\psi_{\mathrm{LTR}}$ = 0.1, respectively. Figure 9 shows the three-dimensional relationship between the *E*, $\psi_{H}$, and $\psi_{\mathrm{HTR}}$. The spherical point in the Figure 9 is the maximum point of the *E*, and the $\psi_{H}$ at this point is significantly higher than that of the $\psi_{\mathrm{HTR}}$. It shows that under the condition of this parameter, the heater has a great influence on the *E*. The best performance of the cycle can be obtained by adjusting the values of the $m_{\mathrm{wf}}$, π, and *x*_p_ and further optimizing the $\psi_{H}$, $\psi_{\mathrm{HTR}}$, $\psi_{L}$, and $\psi_{\mathrm{LTR}}$.

### 3.2. Performance Optimization

This section will aim to maximize *E* under the constraint that the *U*_T_ is a fixed value. The optimization is carried out with the $m_{\mathrm{wf}}$, *x*_p_, π, $\psi_{H}$, $\psi_{\mathrm{HTR}}$, and $\psi_{\mathrm{LTR}}$ as the optimization variables, and the specific optimization process is shown in Figure 10.

The parameter constraints are shown as follows:(24)$$\left\{\begin{array}{l} 60\leq m_{\mathrm{wf}}\leq 140\left(\mathrm{kg}\cdot s^{-1}\right)\\ 2\leq \pi \leq 7.5\\ 0.45\leq x_{p}\leq 0.9\\ 0.01\leq \psi_{\mathrm{LTR}}\leq 0.4\\ 0.05\leq \psi_{\mathrm{HTR}}\leq 0.5\\ 0.05\leq \psi_{H}\leq 0.5 \end{array}\right.$$

At the same time, it is best to keep the *T*_1_ above the critical temperature, so an additional constraint is given by
(25)$$T_{1}\geq 304.42K$$

The performance optimization of the RCSCBC includes six optimization variables, including *m*_wf_, π, *x*_p_ and $\psi_{\mathrm{LTR}}$, $\psi_{H}$, $\psi_{\mathrm{HTR}}$. Through the traditional solution method, there will be a problem of a large amount of calculation, and the @fsolve function is used in the calculation process to rely heavily on the initial value. The interruption of the calculation program caused by calling the REFPROP physical property library to report an error has added more difficulties to the research work. Using neural network prediction can increase the calculation speed, eliminate the need to calculate the ecological function by solving the nonlinear equation system, and get rid of the program’s dependence on the calculation of the initial value. In the case of many optimization variables, neural network prediction is an effective method to simplify the calculation steps.

Therefore, this section introduces neural network prediction in the optimization of the cycle’s single-objective performance.

The method of using the neural network is relatively simple. As long as the corresponding optimization variables are input into the neural network, the corresponding performance target can be obtained. A neural network can be thought of as a very convenient function. In the optimization process, it is only necessary to specify the value range of the corresponding optimization variables and then use the global search algorithm @globalsearch to call the neural network to obtain the optimal ecological function under different conditions.

Among them, constructing a neural network requires correct sample data to construct a mapping function, so its training is carried out on the basis of computational data. By constructing a neural network with a large sample amount of data, the target value of the cycle can be predicted only by entering the corresponding parameter values. Additionally, the more sample data, the closer the final trained model is to the real function. The parameter settings for training the neural network are shown in Table 2.

Neural networks are not just the starting point for optimization algorithms. When there are few optimization variables, the calculation process is relatively simple, but every time one more variable is released, the calculation process becomes complicated. Therefore, a global search algorithm is used for each optimization to call the neural network for optimization.

The construction of the neural network model requires sample points, and each optimization variable is used as the input value; ecological function is the output value. The value range of each variable is shown in Equation (24). There were 6227 random sample points within the calculation range obtained by calculation, and these data are used as samples to train the neural network prediction model.

To verify the reliability of neural network prediction model, it is necessary to re-collect 36 sets of data and compare the calculated values with the predicted values. The re-collected 36 sets of test data cannot be any of the sample data. The comparison results are shown in Figure 11. According to Figure 11, neural network prediction is reliable.

Let π = 3, *x*_p_ = 0.8, *m*_wf_ = 140 kg·s^−1^, and $\psi_{\mathrm{LTR}}$ = 0.1. Figure 12 shows the variation trend of *E* and its corresponding $\psi_{H}$ and $\psi_{L}$ with $\psi_{\mathrm{HTR}}$. From Figure 12, with the increase of $\psi_{\mathrm{HTR}}$, *E* first increases and then decreases, $\psi_{H}$ gradually decreases, and $\psi_{L}$ increases first and then decreases. When $\psi_{\mathrm{HTR}}$ = 0.38, $\psi_{H}$ = 0.25, and $\psi_{L}$ = 0.27, *E* reaches the maximum value. The reason for the difference from the optimal value given in Figure 9 comes from two aspects: one is that there is a certain error in the neural network itself, and the other is that the prediction of the neural network has volatility, but this part of the gap is within an acceptable range.

Let π = 3, *x*_p_ = 0.8, and *m*_wf_ = 140. Figure 13 shows the maximum *E* and the corresponding optimal $\psi_{\mathrm{HTR}}$, $\psi_{H}$, and $\psi_{L}$ under different $\psi_{\mathrm{LTR}}$s. From Figure 13, with the increase of $\psi_{\mathrm{LTR}}$, *E* first increases and then decreases, and its corresponding optimal $\psi_{\mathrm{HTR}}$ has a very obvious downward trend. $\psi_{L}$ first increases and then decreases, but the change is small. Although $\psi_{H}$ also shows a downward trend, the change is relatively small. It shows that under the premise of given other parameters, there is a relationship of mutual influence and restriction between the LTR and HTR. The reasonable distribution of $\psi_{\mathrm{HTR}}$ and $\psi_{\mathrm{LTR}}$ can make the ecological performance of the RCSCBC reach a better level.

Let π = 3 and *x*_p_ = 0.8. Figure 14 shows the *E* of the RCSCBC and its corresponding $\psi_{\mathrm{HTR}}$, $\psi_{H}$, $\psi_{\mathrm{LTR}}$, and $\psi_{L}$ under different $m_{\mathrm{wf}}$. It can be seen from Figure 14 that, given the *x*_p_ and π, the optimal value of the cyclic *E* first increases and then decreases with the increase of the $m_{\mathrm{wf}}$. Additionally, it reaches the maximum value when the $m_{\mathrm{wf}}$ is in the range of 100~110 kg·s^−1^. The changing trends of the four heat conductance distribution ratios with the *m*_wf_ show that $\psi_{H}$ gradually decreases, $\psi_{L}$ gradually increases, and $\psi_{\mathrm{LTR}}$ and $\psi_{\mathrm{HTR}}$ fluctuate up and down within a certain range. When the *m*_wf_ is 70, 80, and 90 kg·s^−1^, there are inflection points in the $\psi_{H}$ and $\psi_{\mathrm{LTR}}$. The main reason for these inflection points is that the *T*_1_ needs to exceed the critical temperature of carbon dioxide.

Taking into account the possibility of too many variables causing the curves to be unclear in the picture, the images after four- and five-time optimizations are no longer given, and the results after four- and five-time optimizations are directly given in Table 3.

Table 3 shows the results obtained by performing one-, two-, three-, four- and five-time optimizations in turn with the goal of maximizing *E*. It can be seen from Table 3 that after the one-time optimization, the *E* can be improved by 12.13% compared with the IDP. After the two-time optimization, the *E* can be improved by 31.52%. After the three-time optimizations, the *E* can be improved by 52.2%. After four-time optimization, the *E* can be improved by 93.26%. After five-time optimization, the *E* can be improved by 96.99%. From the changes of cycle parameters in the one-, two- and three-time optimizations, it can be found that when the $m_{\mathrm{wf}}$ is large, the $\psi_{\mathrm{LTR}}$ is the main factor affecting the *E*. After the three-time optimization, the $m_{\mathrm{wf}}$ corresponding to the optimal value of the *E* decreases, the $\psi_{\mathrm{LTR}}$ decreases, and the $\psi_{\mathrm{HTR}}$ increases. This shows that when the $m_{\mathrm{wf}}$ is small, the HTR is the main factor affecting the *E*. In addition, in the three-time optimization, the improvement of the *E* produces a sudden change. It shows that the heat conductance distribution ratio can make the distribution of cycle performance better, but the $m_{\mathrm{wf}}$ plays a much more important role.

Comparing three-, four- and five-time optimizations, it can be found that after releasing the π and $m_{\mathrm{wf}}$, the parameters corresponding to the maximum *E* have changed significantly, which indicates that the π and $m_{\mathrm{wf}}$ are the main factors affecting the *E*.

In the five-time optimization, the *x*_p_ corresponding to the best *E* reaches 0.9, which is the calculation boundary of the selected parameters, which is influenced by other parameters.

Comparing the results of the five-time optimization with those of the four-time optimization, in addition to the increase of the *x*_p_, the optimal π corresponding to the optimal *E* of the cycle also gradually increases, the $\psi_{H}$ increases, the $\psi_{\mathrm{HTR}}$ increases, the $\psi_{\mathrm{LTR}}$ gradually decreases, and the $\psi_{L}$ increases. The values of all parameters have been re-arranged and changed, which further illustrates that the relationship between the various cycle parameters is also mutual influential.

Table 4 shows the maximum *E* of the RCSCBC and its corresponding optimal parameters under different $m_{\mathrm{wf}}$s. According to Table 4, when the $m_{\mathrm{wf}}$ is small, the π and *x*_p_ corresponding to the optimal value of the *E* are relatively large. When the $m_{\mathrm{wf}}$ is larger, the π and *x*_p_ are smaller. According to the definition of the *x*_p_, the larger the *x*_p_, the smaller $m_{\mathrm{wf}}$ entering the re-compressor. When the *x*_p_ is one, the RCSCBC is transformed into a RSCBC with two regenerators connected in series.

This shows that when the $m_{\mathrm{wf}}$ is small, the *E* of the RSCBC is larger and more advantageous, and when the $m_{\mathrm{wf}}$ is large, the *E* of the RCSCBC is larger, which is more advantageous than the RSCBC. Besides, the optimal π gradually decreases with the increase of $m_{\mathrm{wf}}$ and directly affects the ecological performance of the cycle. In actual engineering, appropriate parameters can be selected for different situations according to the data in Table 3 and Table 4 to achieve the best ecological performance of the cycle.

Comparing the data in Table 4, it can be found that when the $m_{\mathrm{wf}}$ is 65, 75 and 100 kg·s^−1^, there are some inflection points in the values of the *x*_p_ and π.This is due to the influence of the computational boundary, the additional constraint from the *T*_1_, and the influence of the volatility of the neural network prediction. Considering the functionality of the RCSCBC, the *x*_p_ cannot take a value of 1, and according to Equation (24), the *x*_p_ cannot take a value greater than 0.9. Under this premise, when the $m_{\mathrm{wf}}$ is small, the calculation of the optimal value of the *E* is limited.

To sum up, when the $m_{\mathrm{wf}}$ is small, the π, heater, and HTR are the main factors influencing the ecological function. At a higher $m_{\mathrm{wf}}$, the effects of the re-compressor, LTR, and cooler on the *E* increases. According to Table 4, after optimization, the *E* can be improved by at least 53.74% compared with the IDP.

Figure 15 shows the *T*_1_, *T*_H,out_, and the *T*_4_ at each value point in Figure 14. From Figure 15, the *T*_1_ is maintained above 304.42K but has not changed significantly. With the increase of the $m_{\mathrm{wf}}$, both the *T*_H,out_ and *T*_4_ decrease. The temperature of each state point can provide a reference for the stable operation of the cycle and reflect the accuracy of the optimization results to a certain extent.

## 4. Conclusions

In this paper, an FTT model of the RCSCBC was established. First, the performance analysis of the *E* was carried out. Then, on the premise of a certain total heat exchanger inventory, the *E* as optimized with the $m_{\mathrm{wf}}$, π, *x*_p_, $\psi_{H}$, $\psi_{\mathrm{HTR}}$, and $\psi_{\mathrm{LTR}}$ as optimization variables, and the following conclusions were obtained.

(1)The values of $\eta_{t}$ and $\eta_{c}$ reflect the irreversibility of the expansion process and the compression process, and the *E* is used as a thermodynamic index for compromising entropy yield and net power. The irreversibility of the cycle can be measured to a certain extent, that is, there is a correlation between $\eta_{t}$, $\eta_{c}$, and *E*. The higher the $\eta_{c}$ and $\eta_{t}$, the smaller the energy loss caused by the irreversibility of the cycle, and the larger the *E* value.(2)When the $m_{\mathrm{wf}}$ is small, the π, heater and HTR are the main influencing factors on the ecological function. When the $m_{\mathrm{wf}}$ is large, the influences of the re-compressor, LTR, and cooler on the *E* increases. A reasonable adjustment of the distribution ratios of $\psi_{H}$, $\psi_{\mathrm{HTR}}$, $\psi_{\mathrm{LTR}}$, and $\psi_{L}$ can make the cycle performance better, but $m_{\mathrm{wf}}$ plays a much more important role.(3)Each cycle parameter not only affects the performance of the cycle, but also has a mutual influence relationship. The ecological function can be increased by 12.13%, 31.52%, 52.2%, 93.26%, and 96.99% compared with the IDP after one-, two-, three-, four-, and five-time optimizations.

## Figures and Tables

**Figure 1 entropy-24-00732-f001:**
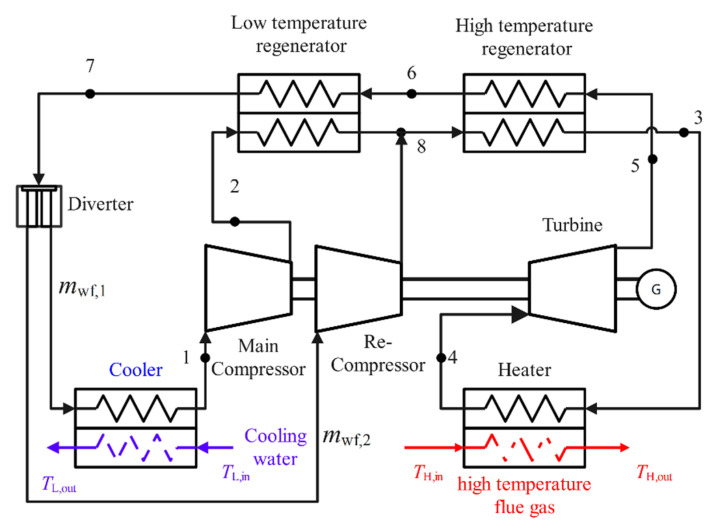
The device diagram of RCSCBC.

**Figure 2 entropy-24-00732-f002:**
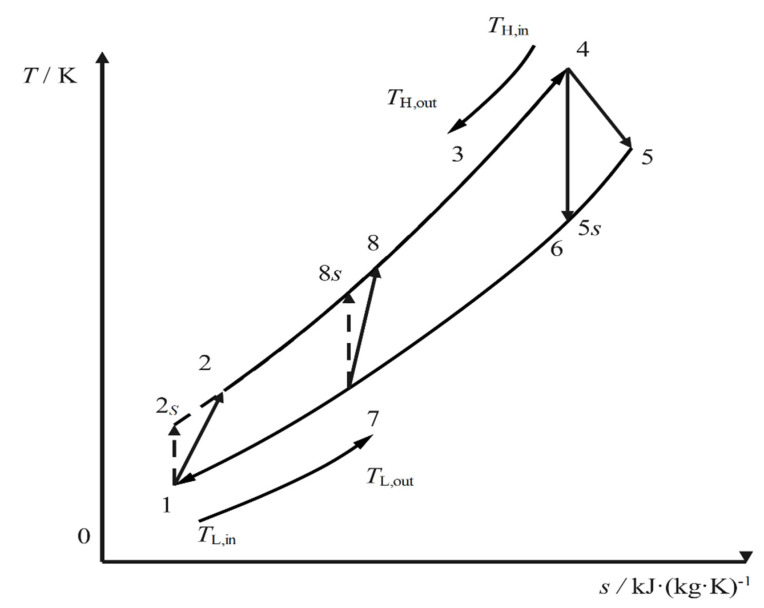
The *T*-*s* diagram of RCSCBC.

**Figure 3 entropy-24-00732-f003:**
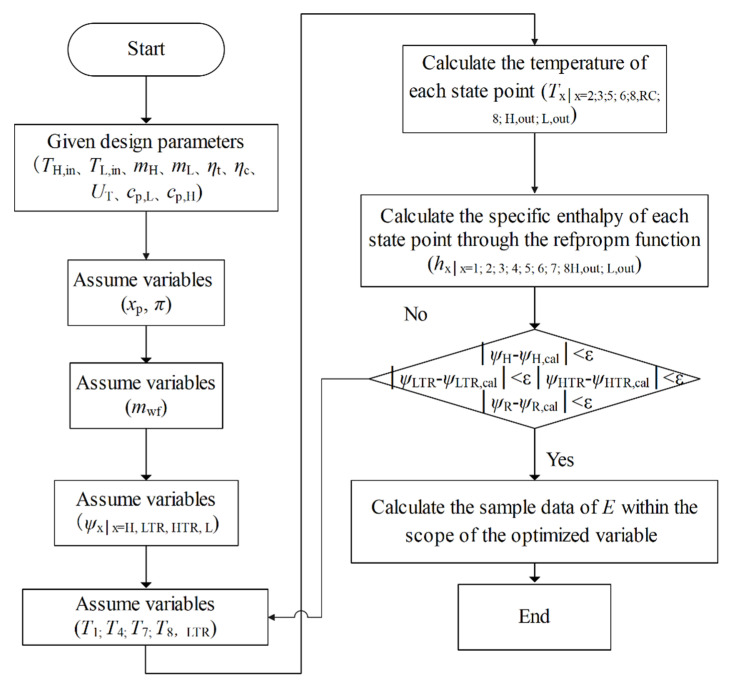
Calculation flow chart.

**Figure 4 entropy-24-00732-f004:**
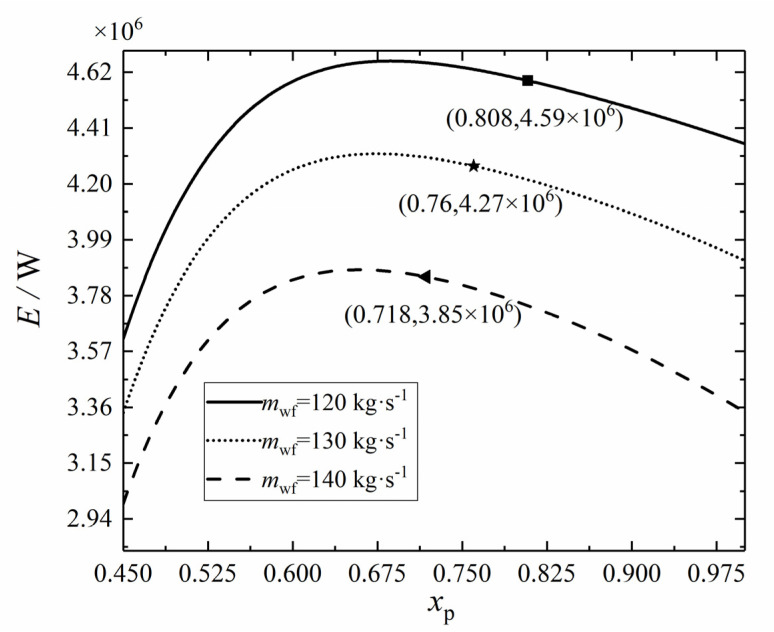
Effect of *m*_wf_ on *E*–*x*_p_ relation.

**Figure 5 entropy-24-00732-f005:**
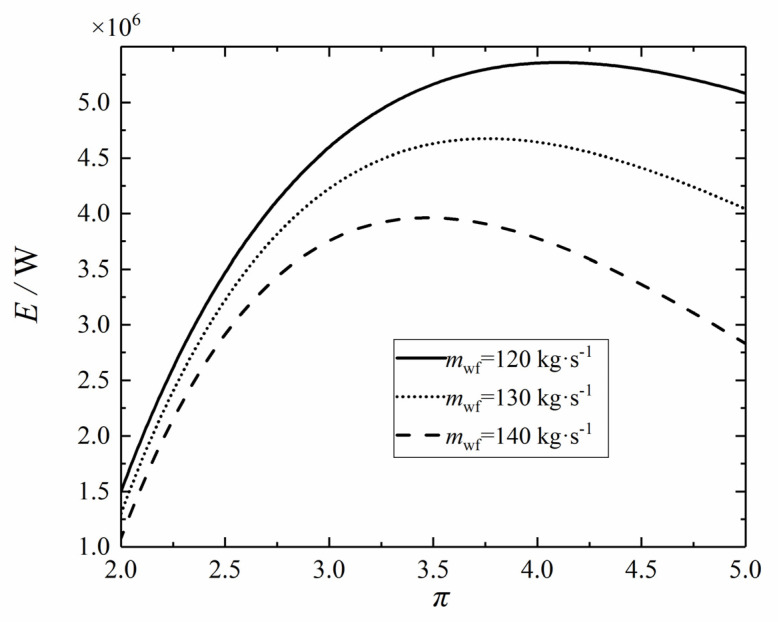
Effect of *m*_wf_ on *E*–*π* relation.

**Figure 6 entropy-24-00732-f006:**
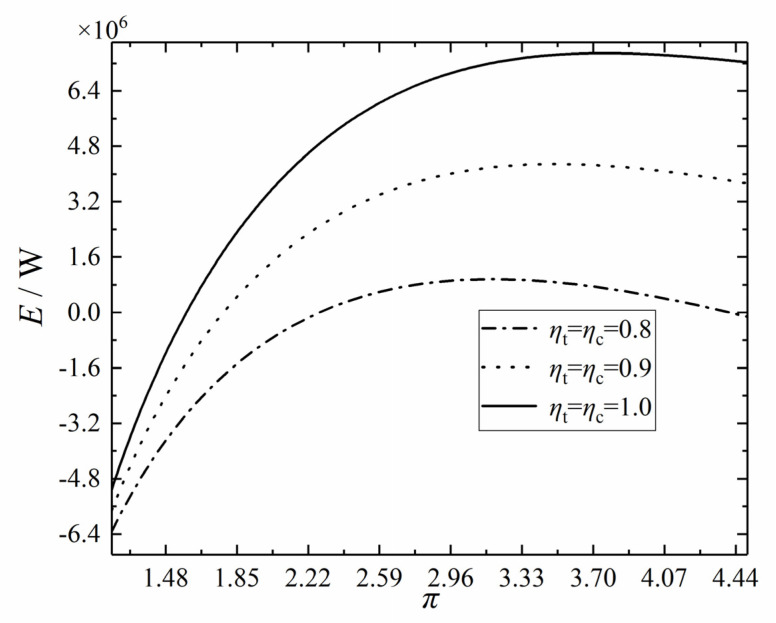
Effect of *η*_t_ and *η*_c_ on *E*–*π* relation.

**Figure 7 entropy-24-00732-f007:**
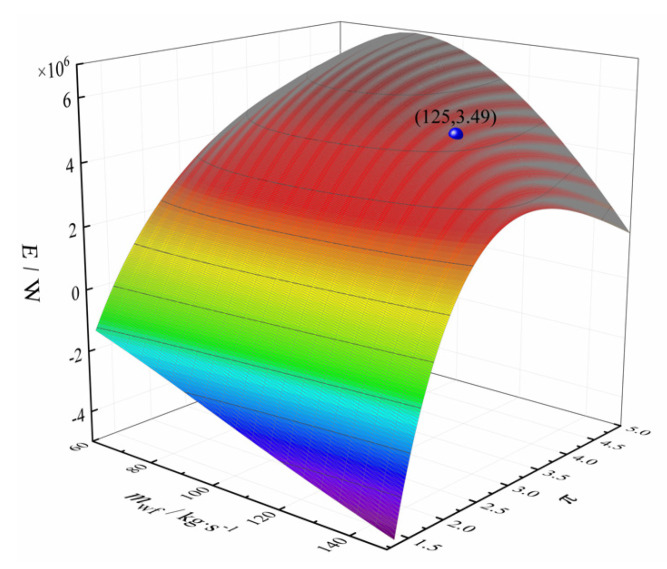
Three-dimensional relationship between *E*, *m*_wf_, and *π*.

**Figure 8 entropy-24-00732-f008:**
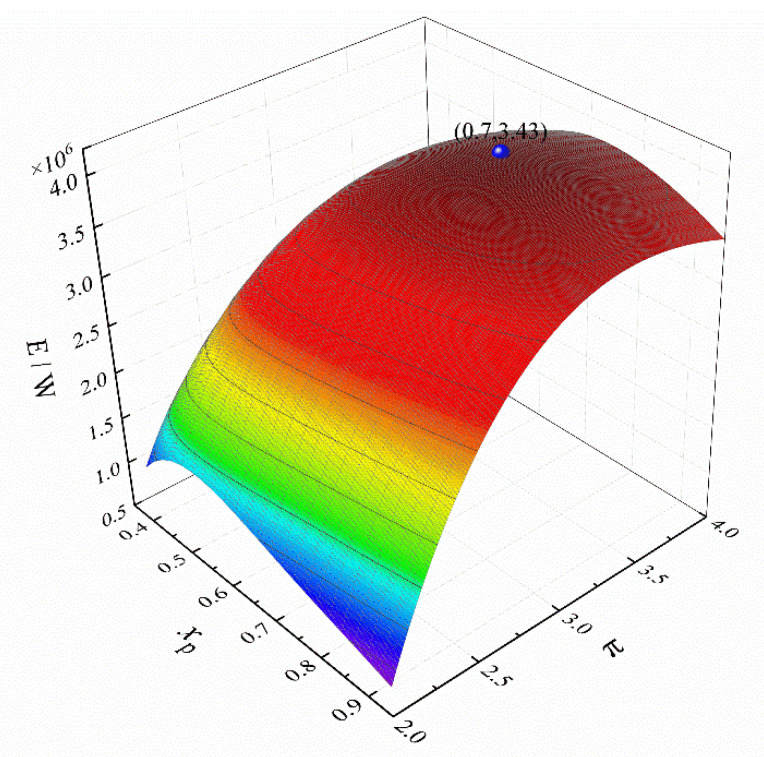
Three-dimensional relationship between *E*, *x*_p_, and *π*.

**Figure 9 entropy-24-00732-f009:**
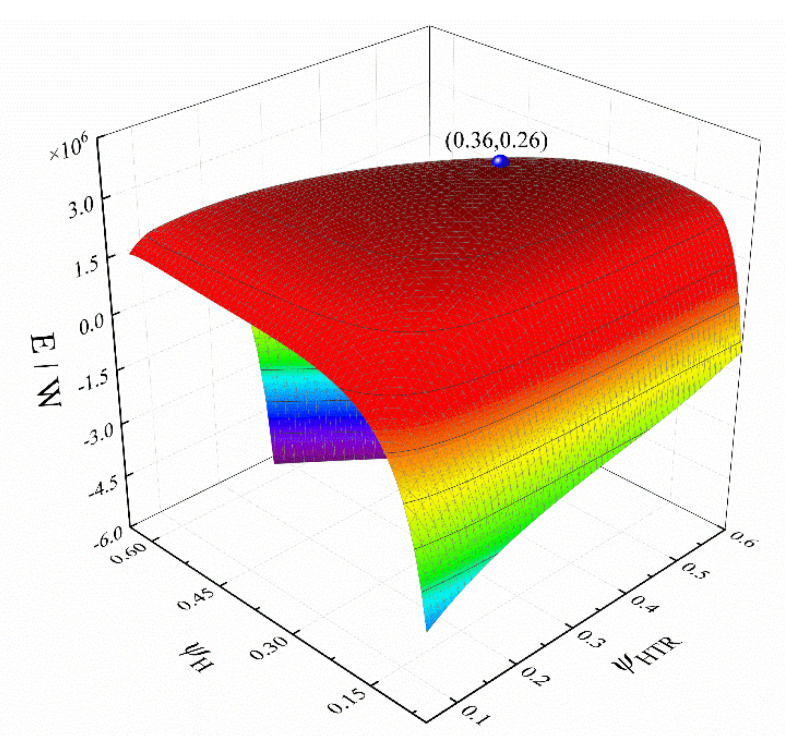
Three-dimensional relationship between *E*, *ψ*_H_, and *ψ*_HTR_.

**Figure 10 entropy-24-00732-f010:**
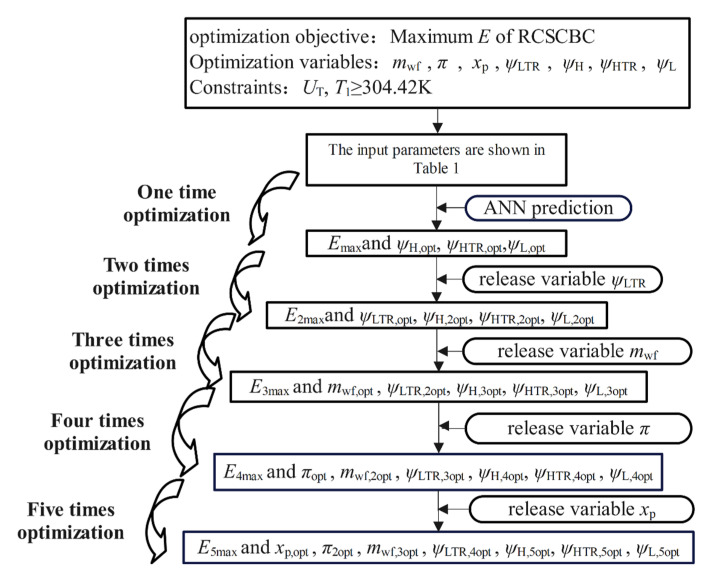
Optimization flow chart of RCSCBC.

**Figure 11 entropy-24-00732-f011:**
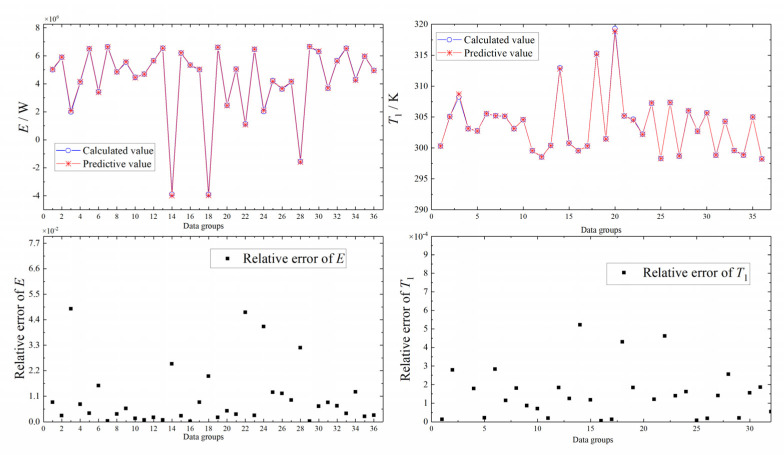
Comparison of predicted and calculated values of *E* and *T*_1_.

**Figure 12 entropy-24-00732-f012:**
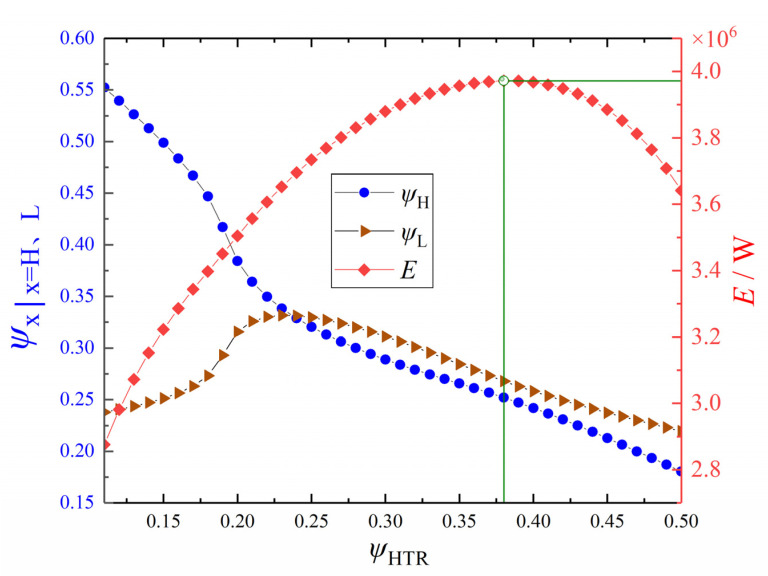
Profiles of *E* and the corresponding *ψ*_H_ and *ψ*_L_ versus *ψ*_HTR_.

**Figure 13 entropy-24-00732-f013:**
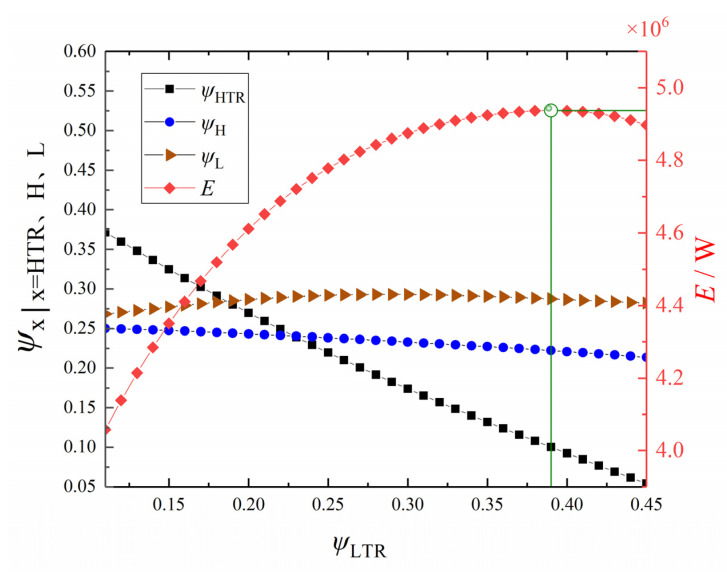
Profiles of *E* and the corresponding *ψ*_HTR_, *ψ*_H_, and *ψ*_L_ versus *ψ*_LTR_.

**Figure 14 entropy-24-00732-f014:**
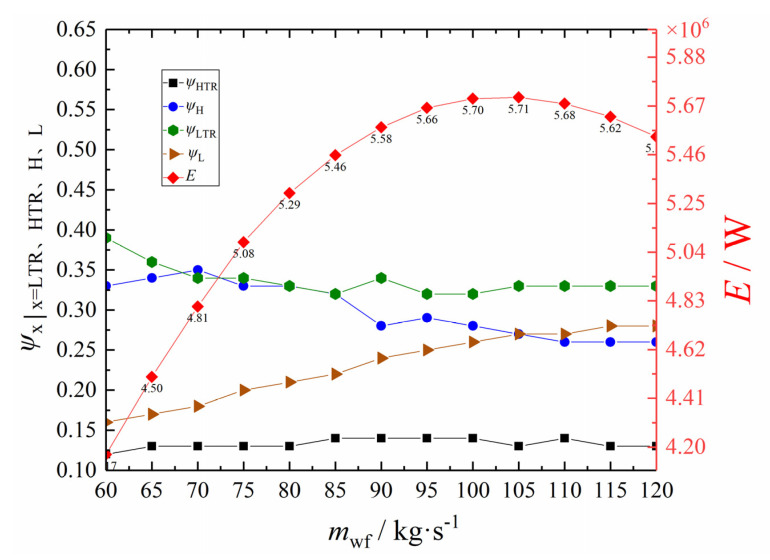
Profiles of *E* and the corresponding *ψ*_PHE_, *ψ*_HE_, *ψ*_R_, and *ψ*_L_ versus *m*_wf_.

**Figure 15 entropy-24-00732-f015:**
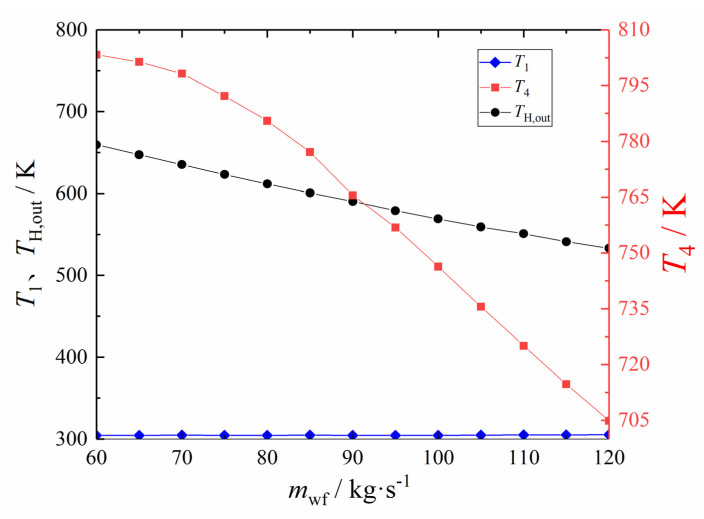
The variation law of *T*_1_, *T*_4_, and *T*_H,out_ corresponding to *E* with *m*_wf_.

**Table 1 entropy-24-00732-t001:** Initial design parameters.

Parameter	Value	Parameter	Value
*T* _H,in_	805.15 K	*η* _t_	0.89
*T* _L,in_	298.15 K	*η* _c,2_	1.0
*m* _H_	89.9 kg·s^−1^	*U* _HTR_	600 kW·K^−1^
*m* _L_	1000 kg·s^−1^	*U* _H_	1200 kW·K^−1^
*m* _wf_	120 kg·s^−1^	*U* _LTR_	300 kW·K^−1^
*x* _p_	0.8	*U* _L_	900 kW·K^−1^
*p* _min_	7.7 MPa	*c_p_* _,L_	4181.3 kJ·(kg·K)^−1^
*p* _max_	20 MPa	*c_p_* _,H_	1103.7 kJ·(kg·K)^−1^
*η* _c_	0.89	-	-

**Table 2 entropy-24-00732-t002:** Parameter settings of neural network model for the RCSCBC.

Parameter Name	*E*	*T* _1_
Samples	6227	6227
Input nodes	6	6
Output	1	1
Hidden layers	2	2
Number of hidden layer nodes layer nodes	30, 15	30, 15
Hidden layer activation function	tansig, purelin	tansig, purelin
Training times	80,000	80,000
Minimum number of confirmation failures	10,000	10,000
Learning rate	120.0	120.0
Minimum training target error	1 × 10^−6^	1 × 10^−7^
Performance function	mse	mse

**Table 3 entropy-24-00732-t003:** Optimization calculation results of *E* based on design point.

Parameters and Objective	Initial Design Point	One-Time Optimization Result	Two-Time Optimization Result	Three-Time Optimization Result	Four-Time Optimization Result	Five-Time Optimization Result
*m*_wf_/kg·s^−1^	140	140	140	105	86.35	85.26
*π*	3	3	3	3	5.02	5.83
*x* _p_	0.8	0.8	0.8	0.8	0.80	0.90
*ψ* _LTR_	0.1	0.1	0.39	0.13	0.28	0.23
*ψ* _HTR_	0.2	0.38	0.1	0.27	0.15	0.16
*ψ* _H_	0.4	0.22	0.22	0.33	0.37	0.39
*ψ* _L_	0.3	0.27	0.29	0.27	0.20	0.22
*E*/×10^6^ W	3.75	3.972	4.937	5.707	7.25	7.387
*δE/*%	-	5.92	31.65	52.2	93.26	96.99

**Table 4 entropy-24-00732-t004:** Calculation results for the case with *E*_max_ as the optimization objective.

Optimization Variables							Objective	Results
*m*_wf_/kg·s^−1^	*π*	*x* _p_	*ψ* _LTR_	*ψ* _HTR_	*ψ* _H_	*ψ* _L_	*E*/×10^6^ W	*δE/*%
60.00	5.76	0.83	0.28	0.21	0.37	0.15	5.998	59.93
65.00	6.55	0.90	0.24	0.29	0.31	0.17	6.512	73.66
70.00	6.15	0.90	0.15	0.20	0.47	0.18	6.844	82.49
75.00	5.88	0.89	0.22	0.22	0.37	0.19	7.127	90.05
80.00	6.00	0.90	0.19	0.17	0.43	0.21	7.305	94.79
85.00	5.83	0.90	0.24	0.18	0.36	0.22	7.350	96.01
90.00	5.67	0.87	0.21	0.16	0.40	0.23	7.286	94.30
95.00	5.04	0.79	0.30	0.15	0.33	0.22	7.205	92.13
100.00	5.19	0.82	0.32	0.15	0.28	0.24	7.029	87.44
105.00	4.47	0.79	0.28	0.17	0.30	0.25	6.900	83.99
110.00	4.60	0.76	0.29	0.18	0.28	0.25	6.712	78.97
115.00	4.20	0.73	0.33	0.15	0.27	0.25	6.606	76.15
120.00	4.04	0.73	0.29	0.18	0.26	0.27	6.371	69.89
125.00	3.84	0.72	0.34	0.15	0.23	0.28	6.212	65.66
130.00	3.71	0.70	0.34	0.15	0.23	0.28	6.006	60.16
135.00	3.58	0.67	0.33	0.18	0.22	0.27	5.765	53.74

## Data Availability

The data that support the findings of this study are available from the corresponding author upon reasonable request.

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
