# Peer review of "Ecological Function Analysis and Optimization of a Recompression S-CO2 Cycle for Gas Turbine Waste Heat Recovery"

_entropy, 2022, doi:10.3390/e24050732_

Round 1

Reviewer 1 Report

The manuscript " Ecological Function Analysis and Optimization of Recompression S-CO2 Cycle for Gas Turbine Waste Heat Recovery " is devoted to study a Brayton cycle with supercritical carbon dioxide model. The ecological function is analyzed for this model; cycle performance was optimized by building an artificial neural network prediction model in the framework of finite-time thermodynamics. It was obtained that the ecological function can be increased by 96.99% compared with the IDP after 5 optimization steps. The results of the work can be useful for optimizing the operating modes of gas turbines and improving energy efficiency of Brayton cycle based devices.

The manuscript is written clearly, well-structured and has a good scientific soundness. I think the manuscript may be published in the Entropy journal only after minor revision after taking into account some of the remarks described below 

  1. It would be better if the Fig 3 was placed in the section 2.
  2. The text between Eqs. (2) and (23) should be formatted correctly.
  3. The abbreviation “WF” should be deciphered.
  4. It is not entirely clear why and how the neural network was used, why its training was conducted on the basis of calculated data? Was it used only to predict the starting points for the optimization algorithm?

Reviewer 2 Report

The reviewed paper presents an interesting analysis of the recompression S-CO2 Brayton cycle. The main novelty of the paper is the development of the finite-time thermodynamics model of the recompression S-CO2 Brayton cycle that takes into account the finite temperature difference heat transfer between the heat source and the working fluid, irreversible compression, expansion, and other irreversibility. Moreover, the artificial neural network-based prediction model is applied in the presented research. The proposed title of the paper represents the content of the manuscript. The abstract states the purpose of the research, the principal results, and major conclusions. The authors presented the scientific background and the state-of-art of the investigated issues in the introduction section and referenced the up-to-date literature. The physical model is properly described. The results and discussion concerning the research methodology and the obtained results are detailed. The conclusions are correctly drawn based on the obtained results.

Therefore, I recommend accepting the paper after miner revision concerning the extension of the nomenclature section as some of the used abbreviations and symbols are not listed there.
